# Development of Radiofluorinated Nicotinamide/Picolinamide Derivatives as Diagnostic Probes for the Detection of Melanoma

**DOI:** 10.3390/ijms22126432

**Published:** 2021-06-16

**Authors:** Yi-Hsuan Lo, Ting-Yu Chang, Chuan-Lin Chen, Ming-Hsien Lin, Hsin-Ell Wang, Chi-Wei Chang, Ren-Shyan Liu, Chun-Yi Wu

**Affiliations:** 1Department of Biomedical Imaging and Radiological Sciences, National Yang Ming Chiao Tung University, Taipei 112, Taiwan; aaa8579@hotmail.com (Y.-H.L.); tingyu870617@gmail.com (T.-Y.C.); clchen2@ym.edu.tw (C.-L.C.); hewang@ym.edu.tw (H.-E.W.); 2Department of Biomedical Imaging and Radiological Sciences, National Yang-Ming University, Taipei 112, Taiwan; 3Department of Nuclear Medicine, Taipei City Hospital Zhongxiao Branch, Taipei 115, Taiwan; cha108009@chgh.org.tw; 4Department of Nuclear Medicine, Cheng Hsin General Hospital, Taipei 112, Taiwan; rsliu@vghtpe.gov.tw; 5Department of Nuclear Medicine, Taipei Veterans General Hospital, Taipei 112, Taiwan; cwchang@vghtpe.gov.tw

**Keywords:** melanoma, radiofluorinated picolinamide–benzamide derivative (^18^F-FPABZA), radiofluorinated nicotinamide–benzamide derivative (^18^F-FNABZA), melanin-targeting probe

## Abstract

Regarding the increased incidence and high mortality rate of malignant melanoma, practical early-detection methods are essential to improve patients’ clinical outcomes. In this study, we successfully prepared novel picolinamide–benzamide (^18^F-FPABZA) and nicotinamide–benzamide (^18^F-FNABZA) conjugates and determined their biological characteristics. The radiochemical yields of ^18^F-FPABZA and ^18^F-FNABZA were 26 ± 5% and 1 ± 0.5%, respectively. ^18^F-FPABZA was more lipophilic (log *P* = 1.48) than ^18^F-FNABZA (log *P* = 0.68). The cellular uptake of ^18^F-FPABZA in melanotic B16F10 cells was relatively higher than that of ^18^F-FNABZA at 15 min post-incubation. However, both radiotracers did not retain in amelanotic A375 cells. The tumor-to-muscle ratios of ^18^F-FPABZA-injected B16F10 tumor-bearing mice increased from 7.6 ± 0.4 at 15 min post-injection (p.i.) to 27.5 ± 16.6 at 3 h p.i., while those administered with ^18^F-FNABZA did not show a similarly dramatic increase throughout the experimental period. The results obtained from biodistribution studies were consistent with those derived from microPET imaging. This study demonstrated that ^18^F-FPABZA is a promising melanin-targeting positron emission tomography (PET) probe for melanotic melanoma.

## 1. Introduction

Malignant melanoma is the most aggressive skin cancer and accounts for approximately 60% of skin cancer deaths [1,2]. The incidence has continuously increased in recent years, especially in North America, Australia, and New Zealand, by 3–8% annually [3]. Clinically, ^18^F-FDG is a standard PET probe to diagnose tumors; however, it could only detect late-stage malignant melanomas due to its non-specificity for melanin. In addition, ^18^F-FDG PET imaging demonstrates relatively high false-positive findings caused by inflammation, pneumonia, infectious processes, and other organs with increased glucose metabolism [4]. The 5-year survival rate for patients with early-stage malignant melanoma receiving surgery (≈92%) was significantly higher than those having distant metastases (≈22.5%) [2]; thus, developing a reliable diagnostic probe for early and accurate detection is crucial for the following treatment.

Melanin is a pigment mainly comprising eumelanin and pheomelanin [3] and is regulated by tyrosinase. One of the characteristics of malignant melanoma cells was the extensive melanin expression due to the elevated tyrosinase activity [4,5]. For this reason, melanin is an excellent specific target for the early diagnosis of melanotic melanoma. Several radiofluorinated and radioiodinated benzamide analogues with high melanin affinities have been developed in recent decades. Michelot et al. first synthesized a benzamide analog, ^125^I-iodobenzamide (^125^I-IBZA), having almost equal accumulations in B16 melanoma (6.75 %ID/g) and liver (6.04 %ID/g) at 1 h after intravenous injection [6]. The first clinical trial of ^123^I-N-(2-diethylaminoethyl)4-Iodobenzamide (^123^I-IBZA) was performed in 1993 and demonstrated its diagnostic sensitivity, accuracy, and specificity as 81%, 87%, and 100%, respectively [7]. However, the liver uptake remained high [7]. Garg et al. prepared a radiofluorinated benzamide derivative, N-(2-diethylaminoethyl)-4-^18^F-fluorobenzamide (^18^F-DAFBA, Figure 1), and found that its uptake in B16F1 melanoma and liver was similar to that of ^125^I-IBZA [8]. The nicotinamide derivative, ^18^F-6-fluoro-N-[2-(diethylamino)ethyl]pyridine-3-carbozamide (^18^F-MEL050, Figure 1), has been prepared with high radiochemical yield and demonstrated an apparent tumor uptake (9.4 ± 1.6 %ID/g at 1 h post-injection) and prolonged retention in melanotic melanoma [9,10]. In 2013, Liu et al. reported that ^18^F-5-fluoro-N-(2-(diethylamino)ethyl)picolinamide (^18^F-5-FPN, Figure 1), a picolinamide analog, displayed a superior in vivo stability and excellent tumor uptake. However, the radiochemical yield of ^18^F-5-FPN was only 5–10%, which may restrict its clinical application [11,12]. In fact, the noted tumor uptakes of these benzamide/nicotinamide/picolinamide analogs are usually accompanied by slow hepatobiliary clearance rates, raising concerns about the detection ability for liver metastases.

We previously demonstrated that the in vivo accumulation of ^131^I-labeled nicotinamide–benzamide derivative, ^131^I-iochlonicotinamide (^131^I-ICNA), in B16F10 melanoma and liver was 13.48 ± 1.77 and 4.17 ± 0.76 %ID/g at 1 h after administration, respectively [13], suggesting that the conjugation with a heterocyclic compound can effectively increase the clearance of benzamide from the normal tissues to cause high tumor-to-background contrast. We also reported that both ^131^I-iodofluoropicolinamide benzamide (^131^I-IFPABZA, Figure 1) and ^131^I-iodofluoronicotiamide benzamide (^131^I-IFNABZA, Figure 1) exhibit superior in vivo targeting ability against melanotic B16F10 melanoma [14]. To have a better imaging resolution, this study aims to prepare novel ^18^F-labeled picolinamide–benzamide (^18^F-FPABZA) and nicotinamide–benzamide (^18^F-FNABZA) derivatives and to determine their feasibility as a diagnostic agent for melanin-expressed melanoma.

## 2. Results

### 2.1. The Preparation of ^18^F-FPABZA and ^18^F-FNABZA

The F-18 was labeled to compounds **2a** and **2b** by the nucleophilic substitution (S_N_2) reaction. The radiolabeling efficiency of ^18^F-FPABZA and ^18^F-FNABZA was 65 ± 5% and 10 ± 5%, respectively (Figure 2A,B). After purification, both compounds’ radiochemical purities were greater than 98% (Figure 2C,D). The radiochemical yields of ^18^F-FPABZA and ^18^F-FNABZA were 25 ± 5% and less than 2%, respectively.

### 2.2. Partition Coefficient Determination and In Vitro Stability of ^18^F-FPABZA or ^18^F-FNABZA

The Log *P* values of ^18^F-FPABZA and ^18^F-FNABZA were 1.48 ± 0.04 and 0.78 ± 0.05, respectively, indicating that ^18^F-FPABZA was more hydrophobic than ^18^F-FNABZA. After incubation in either PBS or FBS, the percentage of intact ^18^F-FPABZA remained greater than 98% until 4 h (Figure 3), demonstrating that no defluorination or decomposition occurred.

### 2.3. In Vitro Binding of ^18^F-FPABZA and ^18^F-FNABZA to Melanin

At a concentration of 0.05 mg/mL, the bound ratios of ^18^F-FPABZA and ^18^F-FNABZA rapidly reached a plateau at the initial 30 min post-incubation (Figure 4A). The binding affinity of ^18^F-FPABZA and ^18^F-FNABZA showed a melanin concentration-dependent manner (Figure 4B). The bound ratios of both radiotracers increased from approximately 20% at melanin concentration of 0.1 mg/mL to higher than 80% at a concentration of 0.2 mg/mL after 1 h incubation. However, the bound ratio of ^18^F-FPABZA was higher than that of ^18^F-FNABZA at each time point, suggesting that ^18^F-FPABZA exhibited a superior affinity to melanin when compared to ^18^F-FNABZA.

### 2.4. In Vitro Cellular Uptake Study of ^18^F-FPABZA and ^18^F-FNABZA in Melanoma Cells

The melanotic B16F10 murine melanoma cells and amelanotic A375 human melanoma cells were used to investigate the specific binding of ^18^F-FPABZA and ^18^F-FNABZA to melanin (Figure 5A,B). The cellular uptake (expressed as %AD/10^6^ cells) of ^18^F-FPABZA in B16F10 cells was 7.57 ± 0.77 at 15 min after incubation and then decreased to 5.00 ± 0.29 at 4 h post-incubation, while that of ^18^F-FNABZA increased from 4.81 ± 0.23 at 15 min post-incubation to 7.82 ± 0.52 after 4 h incubation, which was comparable to the peak uptake of ^18^F-FPABZA (7.94 ± 0.86 at 60 min post-incubation). However, the radioactivity of both radiotracers retained in A375 cells was significantly lower than that in melanotic B16F10 cells at all time points. The maximum uptake of ^18^F-FPABZA and ^18^F-FNABZA in A375 cells was 1.43 ± 0.08 and 3.03 ± 0.07, respectively. The highest melanotic-to-amelanotic cells ratio of ^18^F-FPABZA and ^18^F-FNABZA was 7.94 ± 0.86 at 30 min and 7.82 ± 0.52 at 4 h post-incubation, respectively (Figure 5C).

### 2.5. MicroPET Imaging

The images of B16F10 melanoma-bearing mice injected with ^18^F-FPABZA displayed a noted tumor uptake. The T/M of ^18^F-FPABZA-injected B16F10 melanoma-bearing mice increased from 7.6 ± 0.4 at 15 min p.i. to 27.5 ± 16.6 at 3 h p.i. (Figure 6A). In contrast to B16F10 melanoma, only limited ^18^F-FPABZA radioactivity was retained in the A375 tumor at all imaging points (Figure 6A and Appendix A). In both animal models, intense lung accumulation was noticed at the initial time points and then continuously washed out. In fact, the lung uptake was close to the background at 3 h p.i. in the B16F10 melanoma-bearing mice (Figure 6A). The T/M of A375 xenograft-bearing mice administered with ^18^F-FPABZA was around 1 throughout the experiment period, suggesting that no specific uptake occurred in the amelanotic tumor. Interestingly, although the B16F10 tumor uptake of ^18^F-FNABZA slightly increased with time, it was not comparable to that of ^18^F-FPABZA. The T/M raised from 2.8 ± 0.7 at 15 min p.i. to 5.1 ± 2.9 at 3 h p.i. (Figure 6B). Except for melanotic tumor, the eye was also a high melanin-abundant tissue in C57BL6 mouse, so the accumulation of radiotracers in eyes can be considered an index of specific binding affinity toward melanin. Both ^18^F-FPABZA and ^18^F-FNABZA can clearly delineate the contour of eyes of C57BL6 mice, but the eye uptake of nude mice injected with ^18^F-FPABZA was not observed (Appendix A).

### 2.6. Biodistribution Study

Based on the results obtained from microPET imaging, only the biodistribution studies of ^18^F-FPABZA in B16F10 melanoma-bearing mice were performed (Table 1). At 15 min p.i., noted radioactivity was retained in the lung (8.22 ± 1.87 %ID/g), spleen (11.73 ± 1.37 %ID/g), pancreas (12.80 ± 5.60 %ID/g), and kidney (15.57 ± 1.71 %ID/g), which was accompanied with relatively low blood radioactivity (3.7 ± 0.8 %ID/g), suggesting that the majority of ^18^F-FPABZA rapidly distributed to the whole body from blood. Most of the radioactivity in the normal organ was continuously washed out, indicating that these organs’ accumulation originated from non-specific retention rather than specific uptake. The high radioactivity level in kidney and small intestine implied ^18^F-FPABZA was excreted through both the urinary and intestinal routes. The tumor uptake of ^18^F-FPABZA was 12.32 ± 4.13 %ID/g at 0.25 h p.i., reached the maximum at 1 h p.i. (20.57 ± 2.22 %ID/g), and remained high until 2 h p.i. (16.89 ± 2.32 %ID/g). As a result of prolonged tumor retention, the tumor-to-muscle and tumor-to-blood ratios increased with time and were 6.97 ± 1.96 and 4.40 ± 1.22, 6.66 ± 3.16 and 5.18 ± 1.66, 26.47 ± 3.11 and 19.89 ± 1.19, and 86.57 ± 55.39 and 51.93 ± 18.05, at 0.25, 0.5, 1, and 2 h p.i., respectively. More importantly, the maximum tumor-to-liver ratio was 23.61 ± 9.04 at 2 h p.i., demonstrating that the conjugation with picolinamide may be a solution to the high liver non-specific uptake of benzamide analog. The black eyeballs of C57BL/6 mice also displayed high radioactivity uptake throughout the experiment period. The bony accumulation of ^18^F-FPABZA was 1.84 ± 0.39 %ID/g at 0.25 h p.i. and steadily decreased to 1.40 ± 0.17 %ID/g at 2 h p.i., indicating that no obvious defluorination effect occurred in vivo. In general, the findings derived from biodistribution studies were consistent with those observed in microPET imaging.

## 3. Discussion

Regarding there being no appropriate “specific” agent for the detection of early-stage melanoma until now, several benzamide analogs with high sensitivity and specificity to melanin have been developed, and their biological characteristics have been determined [6,7,9,10,13]. However, the high lipophilicity of these probes generally caused relatively low bioavailability and intense liver accumulation. Our previous works demonstrated that benzamide conjugation with picolinamide or nicotinamide would greatly enhance its hydrophilicity without significantly affecting melanin-targeting ability [13,14]. To improve the imaging resolution of ^131^I-labeled radiotracers, we successfully prepared radiofluorinated fluoropicolinamide–benzamide (^18^F-FPABZA) and fluoronicotinamide–benzamide derivatives (^18^F-FNABZA) for PET imaging in the present study. The radiochemical yield of ^18^F-FPABZA was higher than that of ^18^F-5-FPN [11]. However, the labeling efficiency of ^18^F-FNABZA was low despite using the identical radiofluorination method (≈12%, Figure 2B). This phenomenon may be explained by the different electron densities of pyridine rings between two tracers. Theoretically, the 3-position on the pyridine ring is more electron-deficient than the 2-position, causing a driving force for nucleophilic substitution reaction. 

The melanin-binding assays indicated that the melanin-specific binding of ^18^F-FPABZA and ^18^F-FNABZA was higher than that of ^125^I-IBZA (~80%) but lower than ^131^I-IFPABZA and ^131^I-IFNABZA at the corresponding concentration (0.2 mg/mL) (Figure 3) [6,14]. The cellular uptakes of ^18^F-FPABZA and ^18^F-FNABZA in melanotic B16F10 cells were 5.0- and 3.0-fold higher than in amelanotic A375 cells at 1 h after incubation, respectively, suggesting that melanin was the main target of these radiotracers (Figure 4). Although the accumulation of ^18^F-FPABZA reached the maximum (7.94 ± 0.86 %AD/10^6^ cells) at 0.5 h post-incubation, the value was significantly lower than the peak uptakes of ^131^I-IFPABZA (67.8 ± 0.2) and ^131^I-IFNABZA (62.6 ± 0.2) [14]. The possible explanation may be the difference in lipophilicity between these compounds. ^131^I-IFPABZA and ^131^I-IFNABZA were more hydrophobic than ^18^F-FPABZA and ^18^F-FNABZA, causing radioiodinated compounds more easily to enter into the cytoplasm via diffusion. Pham et al. indicated that radiolabeled probes, having a log *P* value > 1.4, owned superior accumulation in B16F10 tumor [15]. In our previous study, we also reported a positive correlation between the lipophilicity of radiotracers and their tumor uptakes [14]. The amounts of used precursor between radioiodination (≈30 μg) and radiofluorination (≈7 mg) may be another reason to cause different cellular uptakes between these radiotracers. The residual precursor would also bind to melanin and result in a competitive effect.

After the injection of ^18^F-FPABZA into B16F10 melanoma-bearing mice, the prolonged retention of radioactivity in tumor and fast washout in normal tissue resulted in a high tumor-to-background ratio (TBR) and imaging contrast (Figure 6A). To further assess the in vivo specific binding affinity of ^18^F-FPABZA to melanin, the imaging of amelanotic A375 xenograft-bearing BALB/c nude mice was performed. As expected, the radioactivity of ^18^F-FPABZA retained in amelanotic A375 tumors as well as eyes were not significant, suggesting that the main reason for ^18^F-FPABZA long-term tumor accumulation was melanin binding (Figure 6A). On the contrary, ^18^F-FNABZA did not display an apparent tumor accumulation and TBR, unlike the results obtained from in vitro experiments (Figure 6B). Regarding that eyes were still visible at 3 h p.i., ^18^F-FNABZA should own moderate in vivo binding affinity to melanin, suggesting that the melanin affinity is not the primary factor to its limited tumor retention.

As a result of the great binding affinity of ^18^F-FPABZA in imaging experiments, we only assessed the biodistribution study of B16F10 melanoma-bearing mice injected with ^18^F-FPABZA in this study. The tumor uptake continuously elevated from 12.32 ± 4.13 %ID/g at 15 min p.i. to 20.57 ± 2.2 %ID/g at 1 h p.i. The tumor-to-liver ratio of ^18^F-FPABZA was 23.61 ± 9.04 at 2 h p.i. due to insignificant liver accumulation after excretion. Considering that the late-stage melanoma often spreads to liver, these results indicated that ^18^F-FPABZA would be an appropriate probe to detect the metastatic melanoma in liver when compared to other published benzamide analogs, such as ^123^I-IBZA, ^18^F-DAFBA, and ^131^I-ICNA [6,7,8,13].

Uveal melanoma (UM), originating form melanocytes of the uveal, which is the middle layer of the eye and comprises the choroid, ciliary body, and iris, represents around 5% of all melanoma [16]. In fact, except for skin, the eye is the most likely site of melanoma throughout the body. UM can sometimes remain clinical silent for many years. Unfortunately, half of patients with UM develop distant metastases, especially in the liver, leading to poor prognosis [17]. Regarding the difficulties of biopsies to ocular melanoma, eye ultrasound, imaging of the blood vessels, and optical coherence tomography (OCT) have been applied to detect ocular melanoma. However, it is still difficult to differentiate the malignant melanoma from the benign one by using these modalities. In 1998, Bacin et al. found that a benzamide analogue, ^123^I-BZA, was suitable for the diagnosis of ocular melanoma in a phase II clinical trial recruiting 48 patients with a suspicion of ocular melanoma [18]. As mentioned before, ^18^F-FPABZA has a similar lipophilicity and relatively low liver uptake when compared with ^123^I-BZA, suggesting it could be a potential probe to detect UM and the metastases in the liver. In addition, the genetic background is somewhat different from that of cutaneous melanoma despite a similar morphology between these two types of melanoma being observed [17,19,20]. For example, the expression of P16INK4a, which is related to the pigmentation, is a distinct feature in UM and can be a targeting mechanism for ^18^F-FPABZA [17]. Further studies are warranted to determine if the PET imaging with appropriate probes can discriminate between the different types of melanoma.

## 4. Materials and Methods

### 4.1. Reagents and Instruments

All reagents and solvents were purchased from commercial vendors and used without purification. 4-Amino-2-methoxybenzoic acid (97%), N,N-diisopropylethylamine (DIPEA, >98%), N,N-diethylethylenediamine (DEDA, >98%), ammonium hydroxide (NH_4_OH), iodine monochloride, and trifluoroacetic acid (TFA, 99%) were purchased from ACROS Organics (Morris, NJ, USA). 1-Ethyl-3-(3-dimethylaminopropyl)carbodiimide (EDC) was obtained from Biosynth Carbosynth (Berkshire, UK). 1-Hydroxybenzotriazole hydrate (HOBt) and 5-fluoropyridine-2-carboxylic acid (98%) were purchased from AK Scientific, Inc. (Union City, CA, USA). 6-Fluoronicotinic acid (98%) was purchased from Accela ChemBio Co., Ltd. (Shanghai, China). Magnesium sulfate (MgSO_4_), potassium carbonate (K_2_CO_3_), and thionyl chloride (SOCl_2_) were obtained from Merck KGaA (Darmstadt, Germany). Thin-layer chromatography (TLC) was performed by silica gel F254-coated aluminum-backed plates (Merck KGaA, Darmstadt, Germany). Nuclear magnetic resonance (NMR) spectra were determined by the Bruker 400 UltraShield NMR spectrometer (Bruker, Billerica, MA, USA) at the Instrumentation Resource Center of National Yang-Ming University (Taipei, Taiwan). The radioactivity was measured by the γ-counter (PerkinElmer 1470 Automatic Gamma Scintillation, PerkinElmer Inc., Shelton, CT, USA).

### 4.2. Preparation of ^18^F-FPABZA and ^18^F-FNABZA

#### 4.2.1. Synthesis of 4-Amino-N-(2-(diethylamine)ethyl)-2-methoxybenzamide (**1**)

The synthetic scheme of the precursors for radiofluorination was illustrated in Figure 7. The preparation of compound **1** was based on our previously published method [14]. 

#### 4.2.2. Synthesis of N-(4-((2-Diethylamino)ethyl)carbamoyl-5-methoxyphenyl)-picolinamide (**2a**) and Synthesis of N-(4-((2-diethylamino)ethyl)carbamoyl-5-methoxyphenyl)-nicotinamide (**2b**)

5-Bromopicolinic acid (250 mg, 1.25 mmol) was dissolved in thionyl chloride (2.6 mL) and refluxed for 3 h. After the reaction, the solvent was removed by a vacco system, and the residue was re-dissolved in anhydrous THF (15 mL). To the reaction flask, compound **1** (395 mg, 1.48 mmol) and K_2_CO_3_ (335 mg, 2.42 mmol) were added, and the mixture was allowed to react at room temperature (r.t.) for 16 h. The solvent was evaporated to dryness afterward. The residue was re-dissolved in dichloromethane and extracted with ddH_2_O thrice. The organic layer was collected, dried over MgSO_4_, and concentrated to give the crude product, which was then purified by silica gel column chromatography using CH_2_Cl_2_/MeOH (10:1, *v*/*v*) as eluent to afford 2a (white solid, in 88% yield). The compound 2b was prepared with the same procedure described above, except for using 6-bromonicotinic acid (250 mg, 1.24 mmol) as the starting material. The chemical yield of 2b was about 87% (white solid). Compound 2a: ^1^H-NMR (CDCl_3_, 400 MHz): δ1.05 (t, J = 6.4 Hz, 6H), 2.60 (q, J = 6.8 Hz, 4H), 2.66 (t, J = 6.2 Hz, 2H), 3.56 (t, J = 4.8 Hz, 2H), 4.04 (s, 3H), 7.07 (d, J = 8.0 Hz, 1H), 7.99 (s, 1H), 7.91 (s, 1H), 8.07 (d, J = 8.0 Hz, 1H), 8.17 (d, J = 8.4 Hz, 1H), 8.23, (d, J = 8.4, 1H), 8.44 (s, 1H), 8.69 (s, 1H). ^13^C NMR (CDCl_3_, 100 MHz): δ11.9, 37.4, 46.8, 51.5, 55.9, 102.5, 112.2, 111.7, 117.6, 123.7, 124.7, 132.9, 140.7, 141.3, 147.8, 149.3, 158.5, 161.6, 164.8. Compound 2b: ^1^H-NMR (CDCl_3_, 400 MHz): δ1.03 (t, J = 7.2 Hz, 6H), 2.56 (q, J = 6.8 Hz, 4H), 2.60 (t, J = 7.6 Hz, 2H), 3.42 (t, J = 4.8 Hz, 2H), 3.92 (s, 3H), 7.02 (d, J = 8.4 Hz, 1H), 7.37 (d, J = 7.6, 1H), 7.89 (s, 1H), 7.98 (d, J = 8.4 Hz, 1H), 8.20 (m, J = 8.4, 1H), 8.46 (s, 1H), 8.96 (d, J = 1.6, 1H). ^13^C NMR (CDCl3, 100 MHz): δ11.8, 37.5, 46.7, 51.4, 55.9, 103.5, 112.3, 117.3, 117.6, 124.3, 129.4, 132.4, 138.2, 142.3, 148.9, 154.5, 158.3, 163.6, 165.1.

#### 4.2.3. Radiolabeling of ^18^F-FPABZA (**3a**) and ^18^F-FNABZA (**3b**)

Aqueous ^18^F-HF was produced by SCANDITRONIX cyclotron using ^18^O (p,n) ^18^F reaction and kindly provided by Taipei Veterans General Hospital (Taipei, Taiwan). ^18^F-HF was loaded into a Plus QMA Sep-Pak cartridge (Waters) pre-conditioned with 10 mL of 0.5 M K_2_CO_3_ and 10 mL of ddH_2_O. The radioactivity-trapped QMA cartridge was eluted with 0.8 mL of eluent (15 mg of kryptofix 2.2.2 and 3 mg of K_2_CO_3_ in acetonitrile/H_2_O = 4/1, *v*/*v*). The solvent was evaporated to dryness at 110 °C. To the reaction vial, 1 mL of anhydrous acetonitrile was added as azeotrope to remove residual water. The precursor (**2a** or **2b**, 7.5 mg) dissolved in anhydrous dimethylformamide (1 mL) was added to the vial. The reaction mixture was reacted at 140 °C for 30 min. After cooling, the mixture was loaded into a Plus C18 Sep-Pak cartridge (Waters) pre-conditioned with 10 mL of methanol and 10 mL of ddH_2_O. The cartridge was sequentially eluted with 80 mL of ddH_2_O, 8 mL of 30% ethanol, and 3 mL of absolute ethanol. The collected final elution was evaporated to dryness and then re-dissolved in 5% ethanol, 5% tween 80, and 2% BSA in normal saline. The solution was filtered by a 0.22 μm filter to give the final product (^18^F-FPABZA or ^18^F-FNABZA).

#### 4.2.4. Synthesis of N-(4-((2-(Diethylamino)ethyl)carbamoyl)-3-methoxyphenyl)-5-fluoropicolinamide (**4a**) and N-(4-((2-(diethylamino)ethyl)carbamoyl)-3-methoxyphenyl)-6-fluoronicotinamide (**4b**)

The synthetic protocols for **4a** and **4b** were identical to those for **2a** and **2b** except for using 5-fluoropyridine-2-carboxylic acid and 6-fluoronicotinic acid as the starting materials, respectively. The chemical yields of pure compounds **4a** and **4b** were around 60% and 50%, respectively. Both compounds appeared as yellow solids. Compound **4a**: ^1^H-NMR (CDCl_3_, 400 MHz): δ1.06 (t, J = 6.4 Hz, 6H), 2.60 (q, J = 6.8 Hz, 4H), 2.66 (t, J = 6.2 Hz, 2H), 3.53 (s, 2H), 4.00 (s, 3H), 7.02 (d, J = 8.4 Hz, 1H), 7.57 (m, J = 8, 1H), 7.95 (s, 1H), 8.18 (d, J = 8.4 Hz, 1H), 8.28 (d, J = 4 Hz, 1H), 8.31, (d, J = 4, 1H), 8.41 (d, J = 10.8, 1H). ^13^C NMR (CDCl_3_, 100 MHz): δ11.8, 37.4, 46.9, 51.5, 55.9, 102.4, 111.7, 117.5, 124.3, 124.5, 132.9, 136.6, 136.9, 141.6, 145.7, 158.6, 160.2, 161.3, 162.8, 164.8. Compound 4b: ^1^H-NMR (CDCl_3_, 400MHz): δ1.06 (t, J = 6.4 Hz, 6H), 2.62 (q, J = 6.8 Hz, 4H), 2.68 (t, J = 6.2 Hz, 2H), 3.53 (s, 2H), 3.97 (s, 3H), 6.97 (d, J = 8.4 Hz, 1H), 7.04 (m, J = 8, 1H), 7.86 (s, 1H), 8.13 (d, J = 8.4 Hz, 1H), 8.35 (d, J = 4 Hz, 1H), 8.42, (s, 1H), 8.78 (s, 1H). ^13^C NMR (CDCl_3_, 100 MHz): δ11.7, 37.4, 46.9, 51.5, 55.9, 103.5, 109.6, 109.9, 112.3, 117.5, 132.2, 132.5, 134.1, 141.1, 141.2, 142.1, 147.5, 147.7, 158.4, 163.3, 165.1.

### 4.3. Partition Coefficient Determination of ^18^F-FPABZA or ^18^F-FNABZA 

The partition coefficients of ^18^F-FPABZA and ^18^F-FNABZA were determined by measuring their distribution between 1-octanol and phosphate buffer solution (PBS). Briefly, 2 μCi of either ^18^F-FPABZA or ^18^F-FNABZA was added to a mixture of 1-octanol (1 mL) and PBS (pH 7.4, 1 mL). After being vigorously shaken by vortex for 5 min and centrifuged at 3000 rpm for 5 min, aliquots of 1-octanol (0.5 mL) were taken out and then added to another tube containing 0.5 mL of 1-octanol and 1 mL of PBS. This process was repeated five times. The counting samples (100 µL) of each layer were aspirated, and their radioactivity was assessed by a gamma counter. The partition coefficient was expressed as Log *P* based on the following Equation (1).
(1)$$Log\;P=\frac{radioactivity\;in\;1-octanol}{radioactivity\;in\;PBS}\;$$

### 4.4. In Vitro Stability of ^18^F-FPABZA 

To determine the in vitro stability, 100 μCi of ^18^F-FPABZA was incubated in either 1 mL of PBS (at 4 °C and 37 °C) or 1 mL of fetal bovine serum (FBS) at 37 °C. At designated time points (0.25, 0.5, 1, 2, and 4 h), an aliquot of sample was aspirated for radio thin-layer chromatography (radio-TLC) measurement. The percentage of intact ^18^F-FPABZA was considered as an index of its in vitro stability.

### 4.5. Binding Affinity to Melanin Assays

To investigate the binding affinity to melanin, 20 μCi of either ^18^F-FPABZA or ^18^F-FNABZA was added to each tube containing various concentrations of melanin (0.01 to 0.2 mg/mL). The mixture was incubated at 37 °C for 1 h. After incubation, the mixture was centrifuged at 20,000× *g* for 5 min. The supernatant was collected and then filtered by a 0.22 μm membrane filter to remove residual bound radiotracer. The radioactivity of the pellet and the filtrate was measured by a γ-counter. To evaluate the effect of incubation time on the melanin-binding affinity, 20 μCi of either ^18^F-FPABZA or ^18^F-FNABZA was incubated at 37 °C for different time durations (0.5, 1, 1.5, and 2 h). The binding affinity was calculated as follows (2).
(2)$$\%\;\mathrm{of}\;\mathrm{radiotracer}\;\mathrm{bound}=1-\frac{\mathrm{radioactivity}\;\mathrm{of}\;\mathrm{filtrate}}{\mathrm{total}\;\mathrm{administered}\;\mathrm{radioactivity}}$$

### 4.6. Cellular Uptake Studies

Melanotic B16F10 and amelanotic A375 melanoma cells were obtained from the Bioresource Collection and Research Center (Taiwan). Both cells were cultured in Dulbecco’s modified Eagle high-glucose medium (Gibco Life Sciences) supplemented with 10% FBS at 37 °C in a humidified atmosphere of 5% CO_2_. Cells were seeded in a 6-well plate at a density of 1 × 10^6^ cells/well and cultured in the incubator. After cells reaching 70% confluence, the culture medium was replaced by 3 mL of fresh medium containing ^18^F-FPABZA or ^18^F-FNABZA (1 μCi/mL). At 0.25, 0.5, 1, 2, 4, and 8 h post-incubation, the medium was aspirated, and the adhered cells were washed twice with 0.5 mL of PBS. The medium and washing PBS were collected and added to a counting tube. Trypsin (0.25%, 0.5 mL) was added to each well, and the plate was incubated at 37 °C for 5 min to detach cells. The cell suspension was taken out and added to another counting tube. The plate was rinsed with PBS (0.25 mL) twice. The washing PBS was also added to the counting tube containing cell suspension. The radioactivity of cells and medium was measured with a γ-counter and normalized to the number of viable cells. The cellular uptake of ^18^F-FPABZA or ^18^F-FNABZA was expressed as the percentage of administrated dose per one million cells (%AD/10^6^ cells).

### 4.7. Establishment of Melanoma-Bearing Mouse Models

All animal experiments were approved by the Institutional Animal Care and Use Committee (IACUC) of National Yang-Ming University (IACUC no. 1060604). For the subcutaneous melanoma-bearing mouse models, B16F10 (5 × 10^5^) and A375 (1 × 10^6^) cells were inoculated in the right flank of male C57BL/6 mice and male BALB/c nude mice, respectively. Biological studies were conducted when the tumor burdens reached 100 ± 50 mm^3^.

### 4.8. MicroPET Scanning

MicroPET images were obtained by the Lab-PET system (FLEX Triumph) at Taipei Veterans General Hospital, Taipei, Taiwan. The tumor-bearing mouse was anesthetized with 2% isoflurane (in oxygen) using a vaporizer system in the prone position during the imaging process. The B16F10 melanoma-bearing mice were injected with approximately 300 μCi of ^18^F-FPABZA or ^18^F-FNABZA through the lateral tail vein, while the A375 melanoma-bearing mice were only injected with ^18^F-FPABZA for 2 h dynamic imaging. For quantitative analysis, a region of interest (ROI) was placed on tumor and contralateral side of muscle. The tumor-to-muscle ratio (T/M) was calculated on the basis of counts per pixel in the ROIs.

### 4.9. MicroPET Scanning

B16F10 tumor-bearing mice were injected with ^18^F-FPABZA (100 μCi/mice) through the tail vein. At 0.25, 0.5, 1, and 2 h post-injection (p.i.), the tissues/organs of interest were harvested, weighed, and subjected to radioactivity measurement using a γ-counter. The results were expressed as percentage of injected dose per gram of tissue/organ (%ID/g).

## 5. Conclusions

To our best knowledge, this study is the first to synthesize radiofluorinated picolinamide–benzamide and nicotinamide–benzamide derivatives and evaluate their biological characteristics. The results from in vivo experiments revealed that ^18^F-FPABZA exhibited better melanoma-targeting ability than ^18^F-FNABZA and relatively low liver accumulation compared to other existing benzamide analogs, suggesting that it could be a potential clinical probe for the detection of melanotic melanoma.

## Figures and Tables

**Figure 1 ijms-22-06432-f001:**
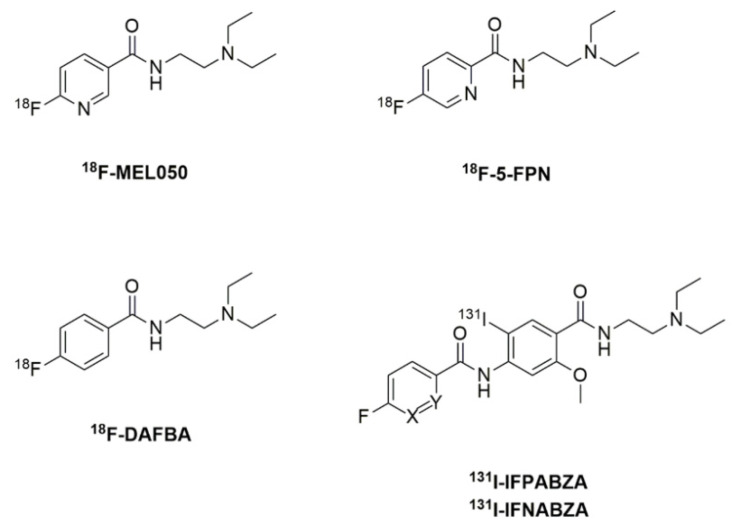
The reported radiotracers for the detection of melanoma.

**Figure 2 ijms-22-06432-f002:**
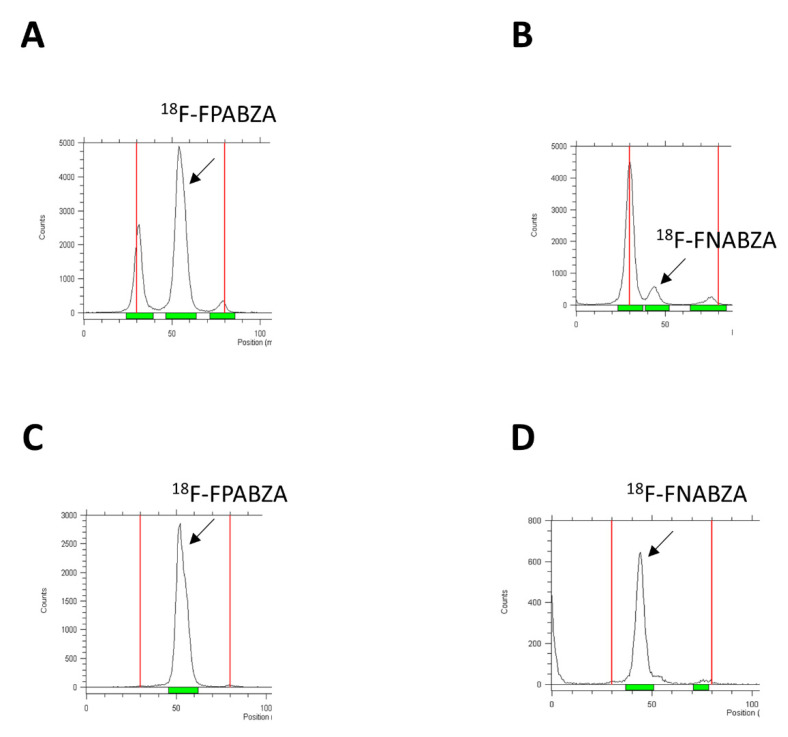
Radio-TLC analysis of (**A**,**B**) the crude products and (**C**,**D**) the purified ^18^F-FPABZA and ^18^F-FNABZA. The radiolabeling efficiency and the radiochemical purity were determined by radio-TLC using MeOH/CH_2_Cl_2_ = 1:3 (*v*/*v*) as the eluting agent.

**Figure 3 ijms-22-06432-f003:**
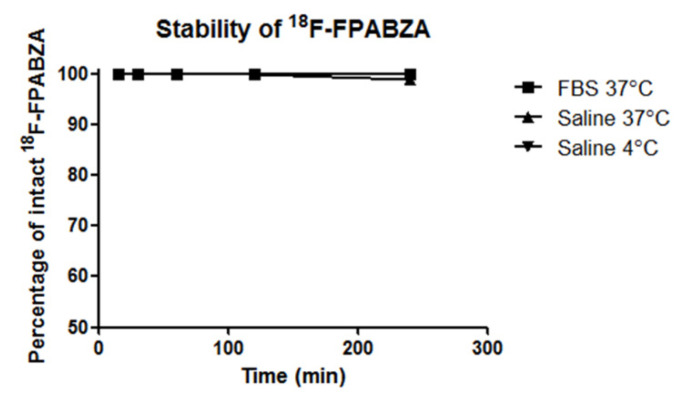
In vitro stability of ^18^F-FPABZA in PBS and fetal bovine serum (FBS).

**Figure 4 ijms-22-06432-f004:**
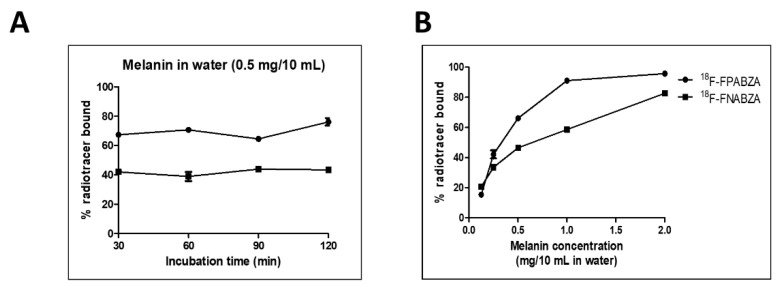
In vitro binding affinity of ^18^F-FPABZA and ^18^F-FNABZA to melanin. (**A**) The bound ratios of ^18^F-FPABZA and ^18^F-FNABZA with melanin suspension (0.5 mg/10 mL) at 37 °C for various incubation durations. (**B**) Binding with various concentrations of melanin at 37 °C for 1 h.

**Figure 5 ijms-22-06432-f005:**
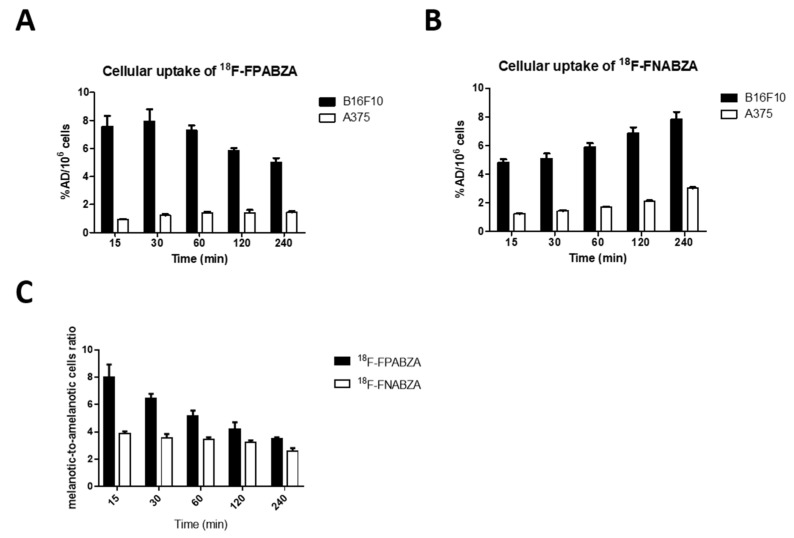
Cellular uptake of (**A**) ^18^F-FPABZA and (**B**) ^18^F-FNABZA in B16F10 murine melanoma and A375 human melanoma cells. (**C**) The melanotic-to-amelanotic cells ratios at each time points. The data are expressed as the mean ± SD (*n* = 5).

**Figure 6 ijms-22-06432-f006:**
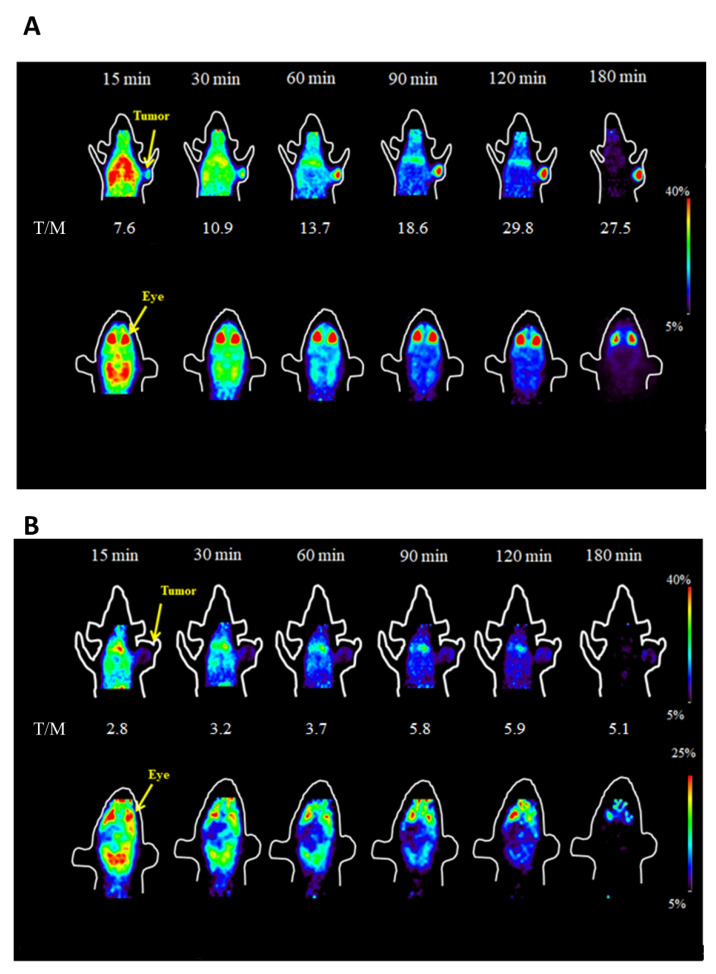
Representative microPET images of B16F10 tumor-bearing C57BL/c mice administered with (**A**) ^18^F-FPABZA and (**B**) ^18^F-FNABZA at 15, 30, 60, 120, and 180 min post-injection. The image slices covered the tumor (top) and eyes (bottom). T/M: tumor-to-muscle ratio.

**Figure 7 ijms-22-06432-f007:**
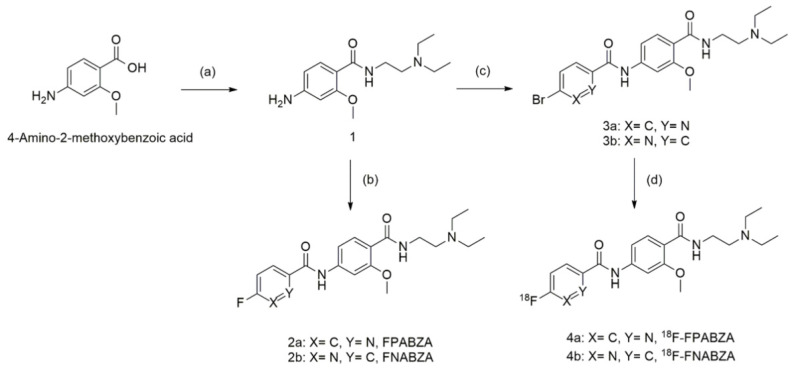
Synthetic scheme of the precursors (2a/2b), radiofluorinated probes (3a/3b), and authentic compounds (4a/4b). Reaction conditions: (a) DEDA, HOBt, EDC, K_2_CO_3_, DIPEA, and DEDA in anhydrous THF, ambient temperature for 16 h; (b) 5-fluoropyridine-2-carboxylic acid/6-fluoronicotinic acid, thionyl chloride, and K_2_CO_3_ in anhydrous THF, ambient temperature for 16 h; (c) 5-bromopicolinic acid/6-bromonicotinic acid, thionyl chloride, and K_2_CO_3_ in anhydrous THF, ambient temperature for 16 h.; (d) ^18^F-HF, K_2_CO_3_, and kryptofix 2.2.2. in anhydrous dimethylformamide, 140 °C for 30 min.

**Table 1 ijms-22-06432-t001:** Radioactivity distribution of ^18^F-FPABZA in B16F10 melanoma-bearing C57BL/6 mice.

Organ	15 min	30 min	60 min	120 min
Blood	3.70 ± 0.80	2.67 ± 0.83	0.95 ± 0.24	0.24 ± 0.17
Heart	4.74 ± 0.80	2.69 ± 0.75	1.00 ± 0.26	0.68 ± 0.69
Lung	8.22 ± 1.87	4.58 ± 1.82	1.82 ± 0.56	1.24 ± 1.34
Liver	5.32 ± 2.04	3.69 ± 1.01	1.63 ± 0.37	1.32 ± 1.00
Stomach	4.95 ± 1.99	3.69 ± 1.35	1.78 ± 0.85	0.78 ± 0.39
Small int.	5.60 ± 0.98	7.35 ± 1.45	5.01 ± 2.64	1.39 ± 0.52
Large int.	4.64 ± 0.65	2.71 ± 0.28	1.91 ± 1.00	1.20 ± 0.32
Spleen	11.73 ± 1.37	5.75 ± 1.85	3.16 ± 1.08	0.54 ± 0.07
Pancreas	12.80 ± 5.60	4.99 ± 1.96	1.93 ± 0.67	0.80 ± 0.22
Bone	1.84 ± 0.39	1.85 ± 0.48	1.40 ± 0.51	1.40 ± 0.17
Muscle	3.32 ± 1.45	2.34 ± 0.81	0.71 ± 0.19	0.44 ± 0.65
Tumor	12.32 ± 4.13	13.21 ± 3.91	20.57 ± 2.22	16.89 ± 2.32
Brain	3.30 ± 0.12	1.26 ± 0.28	0.39 ± 0.12	0.10 ± 0.04
Kidneys	15.57 ± 1.71	10.82 ± 4.29	3.78 ± 1.19	2.39 ± 2.22
Eye ball	20.40 ± 4.81	25.74 ± 3.23	36.91 ± 3.49	33.61 ± 2.14
Urine	103.75 ± 87.59	282.89 ± 115.42	276.62 ± 152.36	90.79 ± 68.19
Fece	6.92 ± 3.79	7.37 ± 1.47	12.18 ± 8.56	12.13 ± 2.86
**Ratios**				
Tumor/muscle	6.97 ± 1.96	6.66 ± 3.16	26.47 ± 3.11	86.57 ± 5.39
Tumor/blood	4.40 ± 1.22	5.18 ± 1.66	19.89 ± 1.19	51.93 ± 18.05
Tumor/liver	2.46 ± 0.38	3.73 ± 1.10	11.03 ± 2.40	23.61 ± 9.04

Results were expressed as the percentage of injected dose per gram of organ/tissue (%ID/g). Each value represented mean ± SD (*n* = 3). Small int., small intestine; large int., large intestine.

## Data Availability

The data presented in this study are available on request from the corresponding author. The data are not publicly available due to ethical issues.

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
