# Peer review of "Development of Radiofluorinated Nicotinamide/Picolinamide Derivatives as Diagnostic Probes for the Detection of Melanoma"

_ijms, 2021, doi:10.3390/ijms22126432_

Round 1

Reviewer 1 Report

The manuscript brings the information about novel probes for diagnosing of malignant melanoma. The manuscript could represent a background for the future experiments in the field of melanoma diagnostics. However I have few questions/notes, that should be clarified or answered:

1. lines 142-143: "The highest melanotic-to-amelanotic cells ratio of 18F-FPABZA and 18F-FNABZA was 7.94±0.86 at 30 min and 7.82±0.52 at 4 h post-incubation, respectively (Figure 5B). " The figure 5B legend does not correlate with this sentence, it does not represent any information about ratios melanotic-to-amelanotic cells. If this ratios should be read from presented graph, than the numbers do not fit.

2. line 231 - "Owing to the limitation of 18F-FDG to detect early-stage melanoma,...." - 18F-FDG as an invasive radioactive test may not have the ambition to detect the early stages of cutaneous melanoma. Authors should rephrase this sentence. Perhaps they mean early detection of metastases, or accidental detection of primary malignant melanoma in an atypical location inside the body, such as in the gastrointestinal tract / abdominal cavity, etc...

3. lines151-153 "In contrast to B16F10 melanoma, only limited 18F-FPABZA radioactivity was retained in the A375 tumor at all Imaging points (Figure 6A)." The figure 6A legend does not correlate with this, but rather supplementary Figure S1. 

4. How authors explain the fact, that on Figure 6A, the radioactivity of eyes is high (red - according to the scale 40% or more) from the beginning (at 0,25h p.i.) and in the tumor it is after 1h p.i. of 18F-FPABZA?

5. What time is needed to completely clear up 18F-FPABZA? 

6. As there is an accumulation of 18FFPABZA also in eye melanin, can't there be a risk of eye damage related to prolonged accumulation of radioactivity in this site? Could the authors provide an opinion on this matter?

7. If the authors claim, that new radiolabels specifically bind to melanin, shouldn't they also be taken up in the brain area of the substantia nigra?

8. I can see few typos: keywords and line 182 "picotinamide" - picolinamide?, line 235 piconamide - picolinamide?, line 436 picoti-amide - picolinamide?

Author Response

Reviewers' Comments

Reviewer #1:

The manuscript brings the information about novel probes for diagnosing of malignant melanoma. The manuscript could represent a background for the future experiments in the field of melanoma diagnostics. However I have few questions/notes, that should be clarified or answered:

  1. lines 142-143: "The highest melanotic-to-amelanotic cells ratio of 18F-FPABZA and 18F-FNABZA was 7.94±0.86 at 30 min and 7.82±0.52 at 4 h post-incubation, respectively (Figure 5B). " The figure 5B legend does not correlate with this sentence, it does not represent any information about ratios melanotic-to-amelanotic cells. If this ratios should be read from presented graph, than the numbers do not fit.

Response: We thank the reviewer for pointing out this and we agree that the melanotic-to-amelanotic ratios should be better demonstrated. We have added the Figure 5C to show these ratios in the revised version.

  1. line 231 - "Owing to the limitation of 18F-FDG to detect early-stage melanoma,...." - 18F-FDG as an invasive radioactive test may not have the ambition to detect the early stages of cutaneous melanoma. Authors should rephrase this sentence. Perhaps they mean early detection of metastases, or accidental detection of primary malignant melanoma in an atypical location inside the body, such as in the gastrointestinal tract / abdominal cavity, etc....

Response: We agree with the reviewer the sentence should be rephrased and have therefore revised the sentences as follows (line 233-235): “Regarding there is no appropriate “specific” agents for the detection of early-stage melanoma till now, several benzamide analogs with high sensitivity and specificity to melanin have been developed and their biological characteristics have been determined.”

  1. lines151-153 "In contrast to B16F10 melanoma, only limited 18F-FPABZA radioactivity was retained in the A375 tumor at all Imaging points (Figure 6A)." The figure 6A legend does not correlate with this, but rather supplementary Figure S1. 

Response: We thank the reviewer for your prudent review of our manuscript and we totally agree that Figure S1 should be added to support the description. We have corrected it and the change have been highlighted in red (line 151).

  1. How authors explain the fact, that on Figure 6A, the radioactivity of eyes is high (red - according to the scale 40% or more) from the beginning (at 0,25h p.i.) and in the tumor it is after 1h p.i. of 18F-FPABZA?

Response: We have demonstrated that both 18F-FPABZA and 18F-FNABZA would specifically bind to melanin in the present study (Figure 4A and 4B) so we think the eyeballs of C57BL/6 mice, containing high amount of melanin, may be the main reason. For the delayed tumor accumulation, we think that high tumor interstitial fluid pressure (IFP) plays a pivotal role in this as it is believed that high IFP of tumor affect the extravasation of the drugs or radiotracers from blood vessel to the lesion.    

  1. What time is needed to completely clear up 18F-FPABZA?

Response: We understand that the clearance rate of 18F-FPABZA is of interest; unfortunately, regarding to the relatively short physical half-life of 18F (109.9 min), all of our imaging and biodistribution studies (Figure 6 and Table 1) were performed at the early time points after intravenous injection. We noticed that the uptake of 18F-FPABZA in normal organs, except for kidney and eyes, decreased with time and almost reached background level at 3 h p.i. Besides, the high radioactivity in urine at 15 min p.i. suggests that 18F-FPABZA was rapidly washed out from blood. The calculated biological half-life of 18F-FPABZA in blood is less than 1 h. Considering the physical half-life of 18F, even assuming 18F-FPABZA would be persistently trapped in the specific organ (having the infinite biological half-life), the radioactivity would drop to 1/4000 of original radioactivity level, which can be regarded as background, at 1 d p.i.

  1. As there is an accumulation of 18F-FPABZA also in eye melanin, can’t there be a risk of eye damage related to prolonged accumulation of radioactivity in this site? Could the authors provide an opinion on this matter?

Response: We agree that the high accumulation of 18F-FPABZA in the eyeballs of C57BL6 mice raises the concern of possible injury in eyes when applied in humans. However, Durairaj et al. indicated that the intraocular melanin contents varies among human, monkey, rabbit, minipig and dog eyes (https://doi.org/10.1016/j.exer.2012.03.004). Labarre et al. and Cachin et al. also found that 123I-IBZA2, a specific melanin-targeting probe, would be significantly stayed in the eyeballs of C57BL6 mice (DOI: 10.1007/s002590050416) while not apparently retained in humans’ eyes (DOI: 10.2967/jnumed.113.123554). These findings suggest that 18F-FPABZA as a diagnostic radiotracer should not cause severe damages to eyes in the clinical applications.

  1. If the authors claim, that new radiolabels specifically bind to melanin, shouldn’t they also be taken up in the brain area of the substantia nigra?-

Response: We thank the reviewer to raise this question that we did not address in the manuscript. Our biodistribution studies showed a low brain uptake of 18F-FPABZA, which was lower than the radioactivity level in blood at all time points (Table 1), suggesting it is difficult to determine the accumulation level of 18F-FPABZA in substantia nigra. This low-level brain uptake may be originated from the ability of compound to cross the blood-brain barrier (BBB). Another issue is that we did not verify the binding affinity of 18F-FPABZA to various neuromelanin. Therefore, we cannot provide a definite answer on this right now. We will perform the related experiments in further studies once we discover a compound having high specific brain uptake.

  1. I can see few typos: keywords and line 182 “picotinamide” – picolinamide?, line 235 piconamide – picolinamide?, line 436 picoti-amide – picolinamide?

Response: We thank the reviewer for the careful review of our manuscript. We have corrected all the typos, located in keywords, lines 184, 238, and 456, and the changes have been highlighted in red. 

Reviewer 2 Report

This manuscript sounds great and it is absolutely well written. The experiments are well conducted and the result clearly presented. I have only a suggestion.

Might this technique discriminate between the different types of melanoma? (uveal vs cutaneous vs mucosal) In particular uveal melanoma (UM) is characterized by an indolent biological history with late onset of liver metastases after 10-15 years from the diagnosis. Authors should include into the text a brief discussion about the potential possibility to use this technique for the diagnosis of all subsets of melanoma. Use these doi for details about UM: 10.3389/fonc.2020.589849 and 10.3389/fonc.2020.562074.

Author Response

This manuscript sounds great and it is absolutely well written. The experiments are well conducted and the result clearly presented. I have only a suggestion.

We thank the reviewer for the kind comments.

  1. Might this technique discriminate between the different types of melanoma? (uveal vs cutaneous vs mucosal) In particular uveal melanoma (UM) is characterized by an indolent biological history with late onset of liver metastases after 10-15 years from the diagnosis. Authors should include into the text a brief discussion about the potential possibility to use this technique for the diagnosis of all subsets of melanoma. Use these doi for details about UM: 10.3389/fonc.2020.589849 and 10.3389/fonc.2020.562074.

Response: We thank the reviewer for pointing out this and agree with the reviewer this information should be added and have therefore added the following sentences to the DICUSSION section (line 288-306): “Uveal melanoma (UM), originating form melanocytes of the uveal, which is the middle layer of the eye and comprises the choroid, ciliary body, and iris, represents around 5% of all melanoma [16]. In fact, except for skin, the eye is the most likely site of melanoma throughout the body. UM can sometimes remain clinical silent for many years. Unfortunately, half of patients with UM develop distant metastases, especially in the liver, leading to poor prognosis [17]. Regarding the difficulties of biopsies to ocular melanoma, eye ultrasound, imaging of the blood vessels and optical coherence tomography (OCT) have been applied to detect ocular melanoma. However, it is still difficult to differentiate the malignant melanoma from the benign one by using these modalities. In 1998, Bacin et al. found a benzamide analogue, 123I-BZA, was suitable for the diagnosis of ocular melanoma in a phase II clinical trial recruiting 48 patients with a suspicion of ocular melanoma [18]. As mentioned before, 18F-FPABZA has a similar lipophilicity and relatively low liver uptake when compared with 123I-BZA, suggesting it could be a potential probe to detect UM and the metastases in the liver. In addition, the genetic background is somewhat different from that of cutaneous melanoma despite a similar morphology between these two types of melanoma was observed [17, 19, 20]. For example, the expression of P16INK4a, which is related to the pigmentation, is a distinct feature in UM and can be a targeting mechanism for 18F-FPABZA [17]. Further studies are warranted to determine if the PET imaging with appropriate probes can discriminate between the different types of melanoma.”

Round 2

Reviewer 1 Report

The manuscript improved significantly and authors addresses all my comments.